# The Optical Properties of Thin Film Alloys of ZnO, TiO$_2$ and ZrO$_2$ with Al$_2$O$_3$ Synthesised Using Atomic Layer Deposition

Natalia Nosidlak [1,*], Janusz Jaglarz [2,*], Andrea Vallati [3], Piotr Dulian [4], Maria Jurzecka-Szymacha [5], Sylwia Gierałtowska [6], Aleksandra Seweryn [6], Łukasz Wachnicki [6], Bartłomiej S. Witkowski [6] and Marek Godlewski [6]

[1] Department of Physics, Cracow University of Technology, 1, Podchorazych Str., 30-084 Cracow, Poland
[2] Faculty of Materials Engineering, Cracow University of Technology, Al. Jana Pawla II 37, 31-864 Cracow, Poland
[3] DIAEE Department of Astronautical, Electrical and Energy Engineering, "Sapienza" University of Rome, Via Eudossiana 18, 00184 Rome, Italy; andrea.vallati@uniroma1.it
[4] Faculty of Chemical Engineering and Technology, Cracow University of Technology, 24, Warszawska Str., 31-155 Cracow, Poland; piotr.dulian@pk.edu.pl
[5] Faculty of Materials Science and Ceramics, AGH University of Science and Technology, Al. Mickiewicza 30, 30-059 Cracow, Poland; maria@agh.edu.pl
[6] Institute of Physics of the Polish Academy of Sciences, Al. Lotnikow 32/46, 02-668 Warsaw, Poland; sgieral@ifpan.edu.pl (S.G.); aseweryn@ifpan.edu.pl (A.S.); lwachn@ifpan.edu.pl (Ł.W.); bwitkow@ifpan.edu.pl (B.S.W.); godlew@ifpan.edu.pl (M.G.)
* Correspondence: natalia.nosidlak@pk.edu.pl (N.N.); pujaglar@cyfronet.pl (J.J.)

**Abstract:** In this work, the results of ellipsometric studies of thin films of broadband oxides (ZnO, TiO$_2$, ZrO$_2$) and broadband oxides doped with Al$_2$O$_3$ (Al$_2$O$_3$–ZnO, Al$_2$O$_3$–TiO$_2$, Al$_2$O$_3$–ZrO$_2$) are presented. All layers have been produced using the atomic layer deposition method. Ellipsometric studies were performed in the wavelength range of 193–1690 nm. Sellmeier and Cauchy models were used to describe the optical properties of the tested layers. Dispersion dependencies of refractive indices were determined for thin layers of broadband oxides on silicon substrates, and then for layers of Al$_2$O$_3$ admixture. The EDX investigations enabled estimation of the composition of the alloys. The Bruggeman effective medium approximation (EMA) model was used to determine the theoretical dependencies of the dispersion refractive indices of the studied alloys. The refractive index values determined using the Bruggeman EMA model are in good agreement with the values determined from the ellipsometric measurements. The doping of thin layers of ZnO, ZrO$_2$ and TiO$_2$ with Al$_2$O$_3$ enables the creation of anti-reflective layers and filters with a specific refractive index.

**Keywords:** ALD deposition; thin dielectric films; optical properties; ellipsometry

## 1. Introduction

In recent years, broadband transition-metal oxides with energy gaps above 3.0 eV have been very popular. The incorporation of nanoparticles into a matrix of transition metal oxides that have a high refractive index *n* greater than 2 can lower its value depending on the degree of doping. This article presents examples of the use of ZnO, ZrO$_2$ and TiO$_2$ oxides alloyed with alumina (Al$_2$O$_3$) in the context of several possible applications, including modified gate oxides and optical filters. Aluminium oxide shows several attractive properties (e.g., large barriers for both electrons and holes) but a low dielectric constant. This is why we decided to verify the possibility of the production of alloys based on this oxide and the ones with larger dielectric constants. As a first step, we selected 50%/50% alloys to verify the possibility and properties of the obtained material.

Thin films of transparent broadband oxides (TBOs) play a significant role in silicon solar cells as antireflection coatings (ARCs) [1]. To reduce the reflection from incoming light, TBO materials should demonstrate low electrical conductivity, as well as low optical

absorption [2]. They possess low reflectance and high transmittance in the UV-VIS-NIR range [3]. Therefore, antireflective TBO films play an essential role in photovoltaics, and they are crucial components in many fields [4,5]. As ARC layers, they can be used on both silicon and glass substrates. In applications of broadband oxide layers as ARCs, they are most often used in two- or three-layer systems.

Currently, there are a number of techniques for achieving TBO synthesis, such as epitaxial growth, hydrothermal synthesis, physical vapour deposition via magnetron sputtering, chemical vapour deposition (CVD), and atomic layer deposition (ALD) [6]. The ALD method is considered as one of the best deposition techniques for producing high-quality thin films due to its self-limited surface reaction behavior and low level of impurity contamination [1,4,7]. For optical applications, transparency and homogeneity of thin films on the substrate are very important [8]. Among all the modern applications for electronic devices, the ALD method became particularly interesting for new applications of oxides such as $TiO_2$, $Al_2O_3$, and ZnO in different areas including biomaterials, photo catalysis and food packaging [5,7,9–11].

Of the various TBO materials, ZnO is one of the most important multifunctional oxides with a wide gap (3.37 eV) and a large excitation binding energy (60 meV) due to its superior optical and electronic properties [7,12–14]. When compared to the other materials, ZnO has emerged as an attractive material for its different kind of properties and flexible application like reflective coatings, anode materials, solar cells, optoelectronic devices, antibacterial specialists and sensor applications [15]. Our results indicate that by introducing Al, the electrical conductivity of ZnO can first be improved. If Al fraction increases above 3%–5% then resistivity of the sample can increase. Material appropriate for ARC can likely be obtained for alloys with large Al fractions.

Second to ZnO is titanium dioxide, $TiO_2$, which has unique characteristics. It is a very versatile transition-metal oxide [16]. It has three different crystallographic phases: brookite, anatase and rutile. $TiO_2$ has a number of attractive properties, such as its wide band gap (3.2 eV) and high refractive index [1,8]. Also, for this material, we decided to test how far we can modify spectral properties by forming an alloy with $Al_2O_3$.

Third, one of the presented TBO materials is $ZrO_2$, which is also one of the most well-studied transition-metal oxides in the optical field due to its very good optical properties [17]. $ZrO_2$ also exhibits a large optical band gap (5.1–7.8 eV), low optical loss and high transparency in the UV, visible and near-infrared region [8]. Many applications of $ZrO_2$ are found in thin-film optics, such as broadband interference filters, and optical and protective coatings [18]. Importantly $ZrO_2$ was also tested (in parallel to $HfO_2$) for gate oxide applications. Mixing with $Al_2O_3$ was interesting to achieve a new material with advantageous properties of components—large barriers for both electrons and holes ($Al_2O_3$) and high dielectric constant ($ZrO_2$).

The application of multilayer systems composed of ZnO, $TiO_2$ and $ZrO_2$ layers can realize not only the full transmission of monochromatic light but also broaden the antireflection spectrum and obtain a better antireflection effect [4]. The production process of bi- and three-layer film was relatively accessible and straightforward [19,20]. For example, bilayer antireflection films consist of two layers with different (low and high) refractive indices.

In the alloys with $Al_2O_3$, the physical properties of the above-mentioned TBO broadband oxide films can be modified. Also, $Al_2O_3$ is widely used for antireflective coatings, protective coatings and transparent electronic device applications [1,11]. An optical band gap is about 6.3 eV and its refractive index in the visible range is about 1.7 [21]. In this paper, we present the optical properties of TBO alloys (ZnO, $TiO_2$ and $ZrO_2$ with $Al_2O_3$). The refractive indices of $Al_2O_3$-TBO systems strongly depend on the presence of $Al_2O_3$ as presented in the paper. Alloying TBO layers with other broadband oxides enables the obtaining of layers suitable for optical filters with optimal optical properties. In this paper, we present the results of the optical study for thin metallic oxides with one and two component layers produced using the ALD method. We would like to show that using the theoretical effective medium approximation (EMA) model, we can obtain refractive

indices of two-component layers that will be consistent with the n values obtained using an actual ellipsometric measurement. Following this path, it will be possible to design a layer with precisely defined refractive index values directly from the theoretical values of the EMA model.

## 2. Materials and Methods

### 2.1. Technology and Samples

We employed the ALD growth method. In this thin film deposition method, the reactants are alternatively introduced to a growth chamber allowing the use of highly reactive precursors, pushing down a growth temperature. The precursors' pulses are separated using inert gas purging. Thus, precursors only meet on the surface of a growing film. The ALD deposition process is, thus, divided into four growth cycles (into four steps): (1) pulse of the first precursor and chemisorptions of the first precursor onto the substrate; (2) purging with an inert gas (in our case, $N_2$) allowing the removal of unreacted precursors; (3) pulse of the second precursor followed by a surface reaction, which produces, in optimal cases, a monolayer of the deposited film; (4) inert gas purge to remove unreacted second precursor and gaseous by-products. The ALD cycle is repeated multiple times until the desired film thickness is obtained. Each of the above-mentioned steps should be optimised to obtain good quality film.

To deposit alloys, we applied the growth procedure used by us to obtain diluted magnetic semiconductors of $Zn_{1-x}Mn_xO$ [22] and $Zn_{1-x}Co_xO$ [23]. In this case, the above-mentioned ALD cycles were repeated m times, followed by n-times repeated cycles in which we changed the first precursor (Zn compound) with a precursor with Mn or Co. The ratio of m to n resulted in alloys of a given composition. At selected growth temperatures, inter-diffusion led to films with a homogeneous composition [22,23].

In the present study, we employed organometallic zinc (diethylzinc ($Zn(CH_3CH)_2$, DEZn (CAS number: 557-20-0), aluminium (trimethylaluminum, TMA (CAS number 75-24-1), zirconia (Tetrakis(dimethylamido)zirconium, TDMAZ, CAS number: 19756-04-8) precursors and an inorganic titanium precursor ($TiCl_4$) (CAS Number: 7550-45-0). All metal precursors were bought from Sigma Aldrich Company (Saint Louis, Missouri, MO, USA) and were tested by us in the deposition processes of selected oxides. Deionised water was used as an oxygen source. Precursor temperatures were low at around 80 °C, the pressure in the growth chamber was a few mbar and the substrate temperature was kept between 100 and 130 °C depending on the process. The deposition process was performed using Savannah-100 Cambridge NanoTech ALD reactor (Cambridge NanoTech, Cambridge, MA, USA). For the growth of alloys, we used a 1:1 ratio of the m to n ALD cycles (see Figure 1). The typical thickness of deposited films was planned to be about 100 nm, but this depended on the growth rates for given elements of an alloy. Films were grown on either Si (reference samples) or glass substrates. No process optimisation was performed. We used deposition parameters standardly used by us in previous ALD studies. Below, we give growth procedures and results for the deposited alloys as follows:

Alloy: $Al_2O_3$–ZnO
1st ALD cycle—$Al_2O_3$: TMA + $H_2O$
2nd ALD cycle—ZnO: DEZ + $H_2O$
Alloy: 10 cycles of TMA + $H_2O$—10 cycles of DEZ + $H_2O$
Alloy: $Al_2O_3$–$TiO_2$
1st ALD cycle—$Al_2O_3$: TMA + $H_2O$
2nd ALD cycle—$TiO_2$: $TiCl_4$ + $H_2O$
Alloy: 10 cycles of TMA + $H_2O$—10 cycles of $TiCl_4$ + $H_2O$
Alloy: $Al_2O_3$–$ZrO_2$
1st ALD cycle—$Al_2O_3$: TMA + $H_2O$
2nd ALD cycle—$ZrO_2$: TDMAZ + $H_2O$
Alloy: 10 cycles of TMA + $H_2O$—10 cycles of TDMAZ + $H_2O$

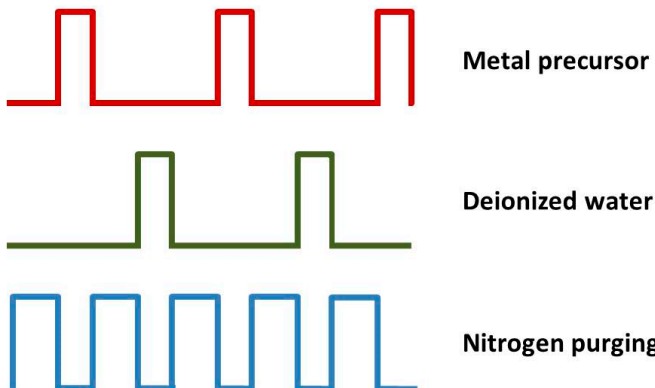

**Figure 1.** Pulse sequence diagram for ALD processes.

ALD cycles were repeated multiple times to obtain the required film thicknesses. Since the use of a second precursor led, in some cases, to the slight etching of the previous layer, we changed the deposition times (up to 100 ms), instead of the 50–60 ms used standardly by us in previous studies. Purging times of a few seconds (typically 4 s) were used, for separating the metal and oxygen precursor pulses. We also used a so-called waiting time (a few seconds) to separate ALD cycles m and n to allow for the interdiffusion. We based this on growth parameters used, e.g., in the reference [24]. As already mentioned, no further process optimisation was performed, since this was the first test to verify the possibility of producing alloys with large metal fractions—intended to be 50%/50%. The data collected in Table 1 indicate that etching, interdiffusion and growth blocking result in alloys with different fractions than initially planned but still with fractions by far larger than those tested by us previously, e.g., ZnMnO [22,23].

**Table 1.** EDX results of the investigated alloys.

| $Al_2O_3$–ZnO | Al% | O% | Zn% | Al/Zn |
|---|---|---|---|---|
| - | 19 | 68 | 13 | 1.46 |
| $Al_2O_3$–$TiO_2$ | Al% | O% | Ti% | Al/Ti |
| - | 12 | 56 | 32 | 0.375 |
| $Al_2O_3$–$ZrO_2$ | Al% | O% | Zr% | Al/Zr |
| - | 12 | 73 | 15 | 0.8 |

*2.2. Characterisation Methods*

The images of the sample cross-section (see Figure 2) were collected using a scanning electron microscope (SEM, Hitachi SU-70, Tokyo, Japan) at an operational voltage of 5 kV with a secondary electron detector.

The EDX analysis was performed using a Thermo Scientific UltraDry silicon drift X-ray detector Thermo Scientific, Waltham, MA, USA) of SEM and Pathfinder X-Ray Microanalysis System Thermo Scientific, Waltham, MA, USA).

The optical properties of the tested samples have been examined using the spectroscopic ellipsometry (SE, Woollam Co. Inc., Lincoln, NE, USA) technique. SE measurements were performed using a Woollam M-2000 ellipsometer (Woollam Co. Inc., Lincoln, NE, USA) with a spectral range of 193 to 1690 nm. SE is a non-destructive and non-contact method that enables the measuring of $\Psi$ and $\Delta$ ellipsometric angles, as well as light depolarisation as a function of wavelength. The use of an appropriate optical model enables the obtaining of information about the tested sample, such as optical constants, thickness and energy bandgap.

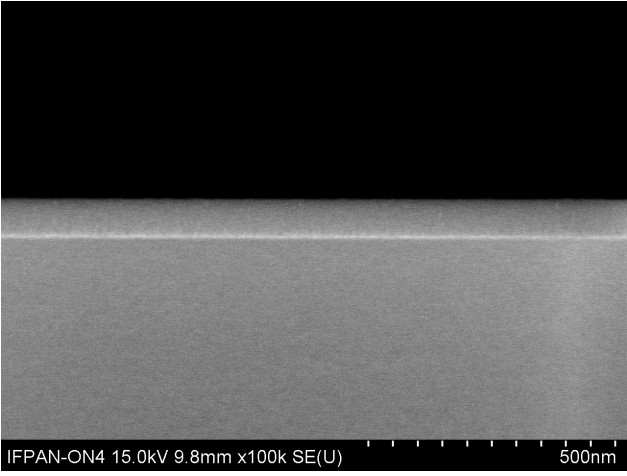

**Figure 2.** Cross section SEM image of an Al$_2$O$_3$–ZnO layer.

The ellipsometric angle $\Delta$ is defined as the phase difference between the *p* and *s* wave function polarisation before and after reflection: the tangent of ellipsometric angle $\Psi$ represents the amplitude ratio for the *p* and *s* wave polarisation also before and after reflection [25]. Both $\Psi$ and $\Delta$ are related to the ratio of Fresnel reflection coefficients $r_p$ and $r_s$ for the light that is polarised parallel (*p*) and perpendicular (*s*) to the plane of incidence. The ratio is defined in Equation (1) as [26]:

$$\rho = \frac{r_p}{r_s} = tan\Psi \; e^{i\Delta} \tag{1}$$

In order to obtain information about the physical properties of the tested thin layer, it is necessary to match the theoretical model to the experimental data.

*2.3. Optical Models*

There are many dispersion relations allowing the obtaining of information about the refractive index and the thickness of the layer examined using ellipsometry measurements. In this paper, we decided to compare the effects of modelling with the Sellmeier model and the extended Cauchy model. Additionally, we present the refractive index dispersion curves obtained from the Bruggeman model, in which the weight percentage of two components in the tested layer is known.

The extended Cauchy model we used is a slightly more developed version of the standard Cauchy model. By adding two additional parameters, *D* and *R*, it enables a more precise description of the refractive index dispersion. The additional parameter *R* causes a rapid decrease in the refractive index values toward the NIR spectral region. The refractive index is given in Equation (2) [27]:

$$n(\lambda) = A + \frac{B}{\lambda^2} + \frac{C}{\lambda^4} + \frac{D}{\lambda^6} - R{\cdot}\lambda^2 \tag{2}$$

where *A*, *B* and *C* are the fit parameters from the standard Cauchy model.

The Sellmeier model () is used to determine the optical constants of transparent dielectric layers. It is a classic description of transparent layers originating from Cauchy's theory [28]. The index of refraction $n_S(\lambda)$ of the Sellmeier model is given in Equation (3) [27]:

$$n(\lambda) = \left( \varepsilon(\infty) + \frac{A\lambda^2}{\lambda^2 - B^2} - E\lambda^2 \right)^{\frac{1}{2}} \tag{3}$$

where $\varepsilon(\infty)$ is an index offset, *A* is the amplitude, *B* is the centre energy and *E* is the position of a pole in the infrared.

The Sellmeier model is more reliable than the Cauchy model, especially for longer wavelengths. The extended Cauchy model, on the other hand, is a semi-empirical model and can be applied to layers with weak absorption.

The dispersion relation of the refractive index of layers composed of two different materials, e.g., $Al_2O_3$ and ZnO can be derived from an appropriate effective medium approximation (EMA). The Bruggeman EMA model (version x) is applied to layers in which the volume fractions of two mixed materials are comparable. The Bruggeman EMA is given using Equation (4) [29]:

$$f_a \frac{\varepsilon_a - \varepsilon_h}{\varepsilon_a + 2\varepsilon_h} + f_b \frac{\varepsilon_b - \varepsilon_h}{\varepsilon_b + 2\varepsilon_h} = 0 \tag{4}$$

where $\varepsilon_a, \varepsilon_b$ are dielectric functions of the constituent materials, $f_a, f_b$ are volume fractions of both mixed materials and $\varepsilon_h$ is the dielectric function of the effective medium itself.

### 3. Results

#### 3.1. SEM and Chemical Structure of Layers

SEM cross section images were measured for all alloys prepared. The one of an $Al_2O_3$–ZnO layer deposited on silicon is shown in Figure 2. Even in this case (ZnO is an n-type material and a small Al dopant lead even to a metallic-like conductivity), an alloy with an increased AL fraction turned out to be resistive. We decided, however, not to coat layers with conducting films, which may not only increase the quality of the picture but also could affect the accuracy of thickness determination, which is crucial for these investigations. Based on the cross-section image, the thickness of all layers was determined. Layer thicknesses determined from SEM were used as starting values in modelling the results of ellipsometric measurements.

SEM images indicate (see the one shown in Figure 2) good uniformity of the obtained alloys, with very flat surfaces. For the reason already mentioned, the presented SEM images are not of high quality due to the insulating properties of the tested layers. Based on previous experiments, we avoided the sputtering of additional layers, e.g., gold or carbon, which is often used to improve the quality of images. In the case of such thin layers, it would only make it difficult to distinguish the tested layer from the sputtered one, affecting the accuracy of thickness determination.

The EDX investigations (see Table 1 and Figure 3) were performed to evaluate alloy properties—for estimation of the composition of the alloys (intended to be 50%) based on ratio n to m ALD pulses. The relevant data are collected in Table 1.

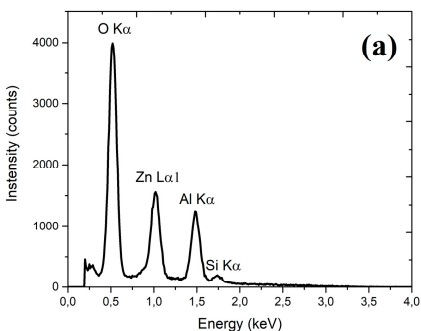 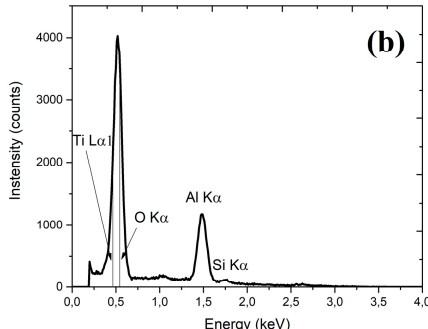 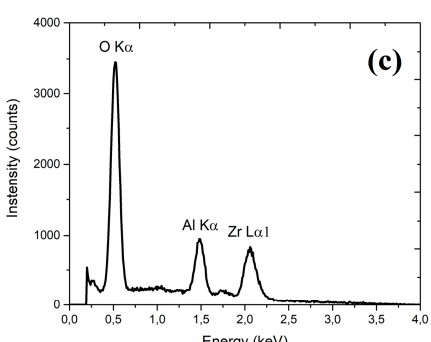

**Figure 3.** Results of the EDX investigations of three types of alloys: $Al_2O_3$–ZnO (**a**), $Al_2O_3$–TiO$_2$ (**b**) and $Al_2O_3$–ZrO$_2$ (**c**).

The authors realize that the EDX technique is not the best method to characterize such thin layers. The beam energy used was selected to be sufficient to generate the signal from the tested elements, but as low as possible so that the generated signal came mainly from the tested layer. In the case of using a higher voltage, a signal could be obtained from higher energy lines, but the depth of signal generation would be much greater, and the signal

would come mainly from the substrate; thus, distorting the results for the layers. Having this in mind, the EDX method was used only to verify that the samples were properly prepared and contained the appropriate compounds, and the percentage results were used only as an estimation for the metals ratio. We also decided not to base on XPS or SIMS investigations. In the case of XPS, we standardly observed a huge signal due to carbon, requiring the etching of samples. For SIMS studies, we usually using much thicker samples.

### 3.2. Optical Studies

SE measurements of $\Psi$ and $\Delta$ angles were performed for all samples at an angle of light incidence on the sample of $70°$. The data were analysed using CompleteEASE 5.15 software. The dispersion relations of the $\Psi$ and $\Delta$, as well as the fitted Sellmeier and Cauchy models for layers on a silicon substrate, are shown in Figure 4.

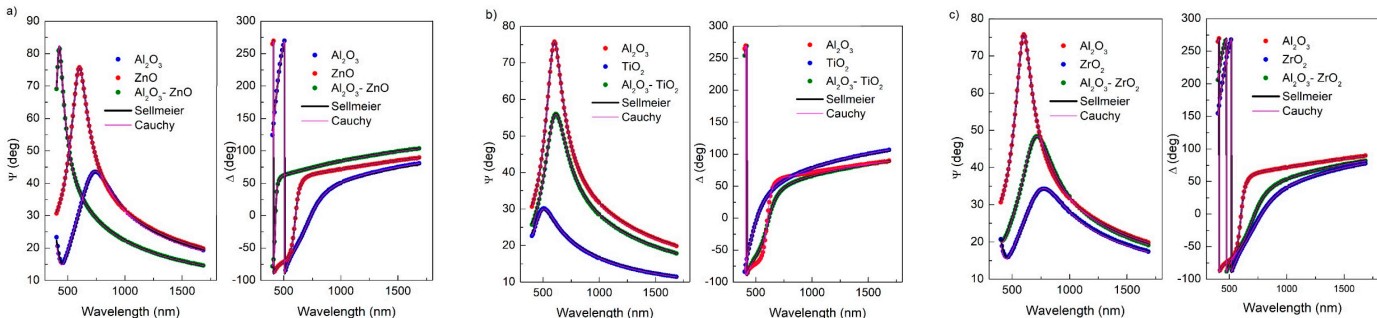

**Figure 4.** Spectral dependence of ellipsometric angles $\Psi$ and $\Delta$ at an incidence angle of $70°$ for: (**a**) $Al_2O_3$–ZnO, (**b**) $Al_2O_3$–$TiO_2$ and (**c**) $Al_2O_3$–$ZrO_2$ layers on a silicon substrate, including Sellmeier and Cauchy models.

As can be seen in Figure 4, both models used in the presented range of wavelengths nearly completely overlap and are indistinguishable in the spectral dependences of ellipsometric angles $\Psi$ and $\Delta$. The mean squared errors (MSE) determined using the Levenberg–Marquardt method were low. For short wavelengths, below 400 nm, the models do not overlap themselves. This is caused by the presence of weak absorption in this region. The spectral range of the J. A. Woollam M-2000 ellipsometer enables measurements from 192 nm, but we have not analysed the results in the UV region due to the occurrence of absorption in this area. We consider only the spectral range where the layers are transparent, due to their potential use as anti-reflection filters.

Dispersion dependencies of refractive indices for layers without $Al_2O_3$ and with the addition of $Al_2O_3$ are presented in Figure 5. It is noticeable that the refractive indices of the layers with the addition of $Al_2O_3$ take intermediate values between those for pure ZnO, $TiO_2$ and $ZrO_2$ layers and the pure $Al_2O_3$ layer, i.e., $n_{ZnO} > n_{mix} > n_{Al_2O_3}$. The resultant value of n is the average value of the dispersion of the refractive index of the oxides constituting the layer. The determined n values for two-component layers obviously depend on the volume ratios of the oxides constituting the layer.

If we assume the occurrence of weak absorption in the VIS region, the experimental results can be fitted with the Cauchy model. Additionally, losses in directional reflection and transmission are due to the scattering of light by surface irregularities and density variations. The extinction coefficient *k* is negligibly small in the presented spectral region; therefore, we have not presented the dispersion relation $k(\lambda)$. The Cauchy and Sellmeier dispersion models in the VIS region give almost the same results, but the differences become visible only in the near infrared.

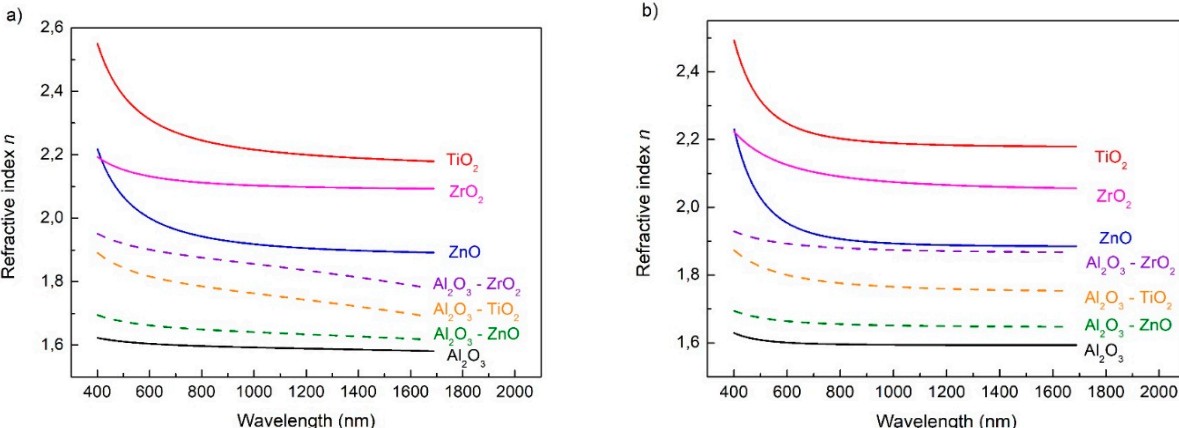

**Figure 5.** Spectral dependencies of refractive indices determined using the (**a**) Sellmeier and (**b**) Cauchy models.

The dispersion dependencies of the refractive index modelled by Sellmeier and Cauchy are presented in Figure 6. The Bruggeman model for mixed oxides $Al_2O_3$–ZnO, $Al_2O_3$–$ZrO_2$ and $Al_2O_3$–$TiO_2$ are also presented in Figure 6.

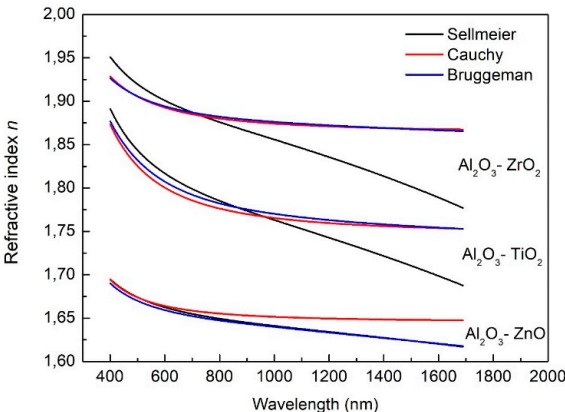

**Figure 6.** Dispersion relations of refractive indices received as a result of modelling with Sellmeier, Cauchy and Bruggeman models.

The refractive index of two-component layers in the EMA theory can be determined using the Maxwell–Garnett EMA or Bruggeman EMA model. For layers with a relatively low content of dopant, the Maxwell–Garnett EMA model is applied, as we have presented in previous work [24]. In this work, the content of $Al_2O_3$ in ZnO, $TiO_2$ and $ZrO_2$ oxides is relatively high The average refractive index for a mixture of two dielectric materials (with known values of refractive indices) with similar content in the layer can be calculated using the Bruggeman EMA model. In Figure 5, it can be seen that the dispersion dependencies of the refractive index for $Al_2O_3$–$ZrO_2$ and $Al_2O_3$–$TiO_2$ obtained as a result of modelling with the Cauchy and Bruggeman models have similar shapes and values. However, in the case of ZnO, the dispersion curve obtained as a result of applying the Bruggeman model corresponds with the curve obtained using the Sellmeier model. Volume percentages used in the Bruggeman EMA model were calculated from the EDX analysis and the results are listed in Table 2.

**Table 2.** Thin-layer thicknesses and refractive indices obtained using the Sellmeier and Cauchy models, refractive indices obtained from Bruggeman EMA and weight percentages of $Al_2O_3$ in mixed layers.

| Thin Layer on Si | | $Al_2O_3$ | $TiO_2$ | $ZrO_2$ | ZnO | $Al_2O_3$–$TiO_2$ | $Al_2O_3$–$ZrO_2$ | $Al_2O_3$–ZnO |
|---|---|---|---|---|---|---|---|---|
| Sellmeier | Thickness (nm) | 116 | 59 | 104 | 108 | 99 | 110 | 76 |
| | $n_{400\,nm}$ | 1.623 | 2.550 | 2.193 | 2.218 | 1.981 | 1.951 | 1.695 |
| | $n_{632\,nm}$ | 1.600 | 2.297 | 2.130 | 1.986 | 1.810 | 1.896 | 1.659 |
| | $n_{900\,nm}$ | 1.593 | 2.228 | 2.107 | 1.928 | 1.774 | 1.866 | 1.645 |
| Cauchy | Thickness (nm) | 116 | 61 | 105 | 111 | 110 | 100 | 76 |
| | $n_{400\,nm}$ | 1.629 | 2.492 | 2.223 | 2.229 | 1.873 | 1.929 | 1.694 |
| | $n_{632\,nm}$ | 1.600 | 2.235 | 2.118 | 1.949 | 1.795 | 1.890 | 1.662 |
| | $n_{900\,nm}$ | 1.595 | 2.194 | 2.081 | 1.898 | 1.770 | 1.877 | 1.653 |
| Bruggeman | $n_{632\,nm}$ | - | - | - | - | 1.802 | 1.891 | 1.657 |
| % of $Al_2O_3$ | | 100 | - | - | - | 32 | 36 | 54 |

The light depolarisation after reflection is determined directly from ellipsometric measurements. The depolarisation coefficient describes the degree of coherence of light in the layer. The depolarisation spectra of pure broadband oxide layers and layers with $Al_2O_3$ are presented in Figure 7.

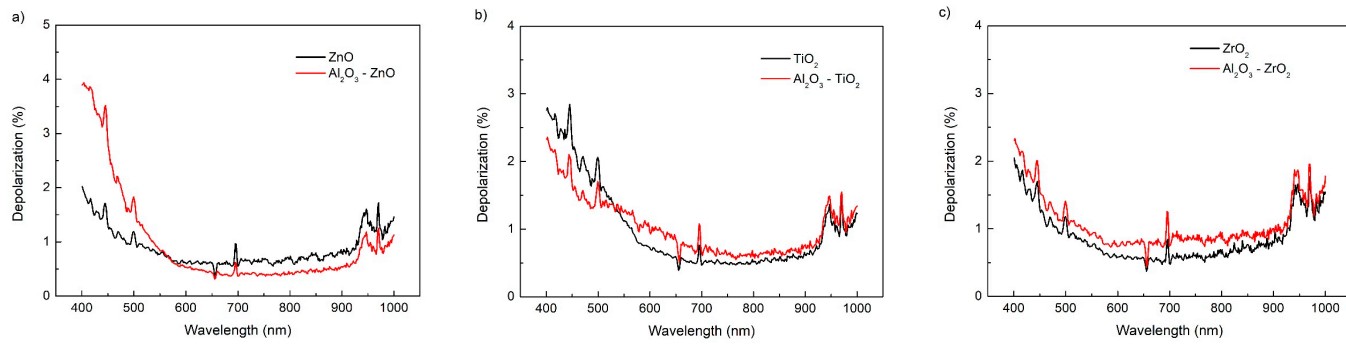

**Figure 7.** Depolarisation spectra of: (**a**) ZnO and $Al_2O_3$–ZnO, (**b**) $TiO_2$ and $Al_2O_3$–$TiO_2$, and (**c**) $ZrO_2$ and $Al_2O_3$–$ZrO_2$.

The depolarisation coefficient describes the quality of the interfaces between the air and the layer and between the layer and the substrate. For wavelengths longer than 1000 nm, the depolarisation coefficient is nearly 0%, while in the range of 400–1000 nm, the depolarisation of the alloy layers is about 0.5% higher than of the layers without the addition of $Al_2O_3$. The exception is ZnO above 550 nm. This means that the polarisation state of the light after reflection is similar for the broadband oxide layers and the alloys. There is no quantitative or qualitative change in the morphology of pure and mixed layers, which can be seen from the similarity in the shape of the depolarisation dispersion curves. A low depolarisation coefficient does not affect the choice of the optical model. The addition of $Al_2O_3$ has a small influence on the depolarisation and, thus, on the diffusion scattering from the layers. The depolarisation coefficient should be taken into account in modelling anti-reflective layers.

Transmission spectra obtained from SE for all layers are shown in Figure 8. The presence of $Al_2O_3$ increases the transparency of mixed layers, which is clearly visible in the range of about 400–1000 nm, i.e., beyond the absorption range.

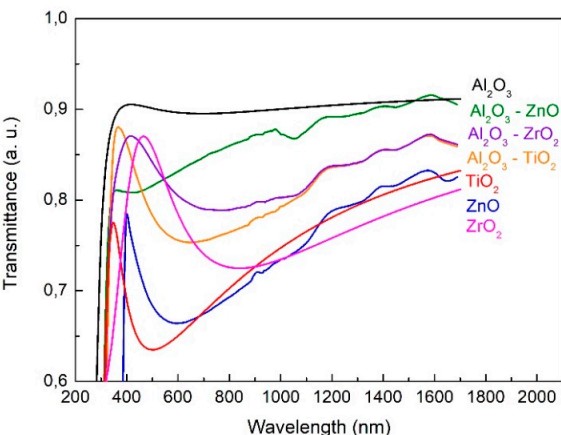

**Figure 8.** Transmission spectra of $Al_2O_3$, ZnO, $TiO_2$, $ZrO_2$, $Al_2O_3$–ZnO, $Al_2O_3$–$TiO_2$, $Al_2O_3$–$ZrO_2$ layers.

Layer thicknesses and refractive indices were determined using the Cauchy and Sellmeier models. The results for three selected wavelengths are presented in Table 2. The determined layer thicknesses and refractive indices values were obtained using the application of both models. A more significant difference in refractive indices values appears for wavelengths longer than 1000 nm. This is due to differences in the applied models.

## 4. Conclusions

In this paper, the results of an optical study of thin layers of ZnO, $ZrO_2$ and $TiO_2$ broadband oxides doped/alloyed with $Al_2O_3$ grown using the atomic layer deposition technique have been presented. Without any optimisation alloys with large metal fractions were investigated. The Sellmeier and Cauchy models were used to determine the dispersion dependencies of the refractive index of single oxide layers and for two-component layers.

The addition of $Al_2O_3$ to broadband oxides (ZnO, $ZrO_2$ and $TiO_2$) reduces the refractive index of the alloy layers compared to the layer without the admixture. By appropriate selection of the volume percentage of $Al_2O_3$ in the layer, it is possible to design layers with new intermediate values of the refractive index. This is a highly important result opening chances for optical properties engineering.

The Bruggeman effective medium approximation model was used to determine the refractive indices without the need to measure thin alloy layers. The refractive index values obtained using Bruggeman EMA modelling correspond to those obtained experimentally. By selecting the appropriate percentages of components ($Al_2O_3$ and broadband oxide) in the layers, we are able to design layers with a specific refractive index.

Using the Fresnel equations, we can design efficient anti-reflective layers. There is no absorption observed in the presented layers; therefore, the layers with content of $Al_2O_3$ have excellent dielectric properties. The depolarisation of the beam reflected from the layers is low, nearly the same as for the undoped broadband oxide layers, which proves their high optical quality. The ability of the controlled influence to changes of the refractive indices of thin layers of ZnO, $ZrO_2$ and $TiO_2$ by doping with $Al_2O_3$ enables the creation of anti-reflective layers and filters.

**Author Contributions:** Conceptualization, N.N., J.J. and M.G.; methodology, P.D.; software, N.N.; validation, P.D., M.G. and B.S.W.; formal analysis, A.S. and S.G.; investigation, A.S. and N.N.; resources M.G. and B.S.W.; data curation, N.N. and S.G.; writing—original draft preparation, N.N., M.J.-S., M.G. and J.J.; writing—review and editing, N.N., J.J. and B.S.W.; visualization, N.N., A.V. and Ł.W.; supervision, M.G.; project administration, J.J. and M.G.; funding acquisition, M.G. and B.S.W. All authors have read and agreed to the published version of the manuscript.

**Funding:** This research was partially funded by the National Centre for Research and Development, grant number POIR.04.01.04-00-0052/20-00.

**Institutional Review Board Statement:** Not applicable.

**Informed Consent Statement:** Not applicable.

**Data Availability Statement:** Not applicable.

**Conflicts of Interest:** The authors declare no conflict of interest.

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
