# Peer review of "The Optical Properties of Thin Film Alloys of ZnO, TiO2 and ZrO2 with Al2O3 Synthesised Using Atomic Layer Deposition"

_coatings, doi:10.3390/coatings13111872_

Round 1

Reviewer 1 Report

Comments and Suggestions for Authors

This paper presents the optical research for thin metallic oxides one and two component layers produced by the atomic layer deposition method. This manuscript presents a systematic and comprehensive work. The content of the manuscript is in line with the research direction of Coatings. There are only a few minor problems with the content in the author's manuscript. The existence of these problems leads me to suggest minor revision.

(1)  In the manuscript, the third section is “Results”. The authors should provide more in-depth discussion.

(2)  Page 3 Line 131-145, I think it is better to show the growth procedures and results using a table or figure.

(3)  Page 3 Line 126, please check the unit of “80 ºC”.

(4)  In this manuscript, the authors use a comma to represent the decimal point. Please confirm if this is canonical.

(5)  Page 3 Line 222, “SEM image indicate good uniformity…” should be “SEM image indicates good uniformity…”.

Author Response

Dear Reviewer,

Thank you very much for your consideration, and we really appreciate the comments. We did not change commas to periods to represent the decimal points  because this convention applies in our country and is respected by other scientific journals. Other suggested changes were made and highlighted in the revised manuscript according to your suggestions.  

Reviewer 2 Report

Comments and Suggestions for Authors

This article is devoted to an important issue - the creation of coatings based on wide-gap semiconductor oxides. The subject of this article, in principle, corresponds to the Coatings MDPI journal

These materials are widely used in various fields of science and technology, such as anti-reflective coatings, as well as transparent contacts for various devices.

The reliability of the results obtained by the authors is also beyond doubt.

However, this article has a significant drawback: it lacks a pronounced element of scientific novelty. The semiconductor systems described by the authors are widely known, and their various properties have been well studied. As the search shows, a number of works on them date back to the 60s. Moreover, ZnO-Al2O3 films have been produced industrially for a long time (AZO material), they are used as conductive contacts for various devices, in particular for thin-film solar cells.

I think that the authors should radically change the concepts of this article.

Author Response

Dear Reviewer,

Thank you very much for your kind and interesting comments. We have expanded the manuscript with additional information about the novelty and the aims of the work. It is true that the material is well known, but the aim of the work was to select a methodology that would allow determining optical properties a priori. We hope that we have achieved this goal. All amendments were made and highlighted in the revised manuscript according to your suggestions.

To answer point 2 specifically we believe that the ratios of Al/Zn, Al/Ti, and Al/Zr are consistent with the estimation as seen in Tables 1 and 2. The results confirming the consistency of the applied model with the percentages in the layers are presented in the tables. The calculated contents based on the EMA model differ from the values determined from the EDX measurement within the measurement error.

The discrepancy between the percentage of Al2O3 in Table 2 and the results from EDX that you mention are the result of the fact that the Table 2 shows the percentage of Al2O3 in the layer, while Table 1 shows the percentage of individual elements in the layer, hence the discrepancies.

Two-component layers can act as anti-reflective layers in multi-layer systems, where it is important to select a layer with a precisely defined refractive index. However, these are not the only applications of this type of oxide layers. Our goal was not to test this type of layers strictly for ARC applications, but to obtain information on how to obtain a specific refractive index of a layer made of two types of oxides. This knowledge will allow, for example, the use of this type of layers as components of Quarter Wavelength Optical Thickness (QWOT) systems.

Our task was to determine the dependence of transmission and refractive indices on the Al2O3 content in metallic oxide layers. We can conclude that the addition of Al2O3 increases the transmission and reduces the reflection in ZnO, TiO2 and ZrO2 oxides layers.

Reviewer 3 Report

Comments and Suggestions for Authors

I have had the privilege to review your manuscript entitled "The optical properties of thin films alloys of ZnO, TiO2 and ZrO2 with Al2O3 synthesised by atomic layer deposition". It is a significant contribution to the study of broadband oxides and their optical properties. Below, I have provided several comments and questions that I believe will help clarify certain aspects of your work and improve the presentation of your findings.

1.     You mentioned that the EDX technique is not the most suitable for characterizing such thin layers. In light of this, it would be beneficial if you could provide depth profiles using Secondary Ion Mass Spectrometry (SIMS) or X-ray Photoelectron Spectroscopy (XPS) to further confirm the alloy compositions.

2.     Referring to Figure 4, you utilized the Sellmeier and Cauchy models to estimate the refractive indices. Given that the alloys were deposited using 10 cycles of growth procedures, can you discuss whether the EDX-determined ratios of Al/Zn, Al/Ti, and Al/Zr are consistent with these estimations?

3.     I noticed that the quality of the cross-section SEM images could be enhanced. It would be imperative to improve these to validate the thickness data presented in Table 2.

4.      Could you provide an explanation for the apparent discrepancy between the percentage of Al2O3 in Table 2 and the results from EDX?

5.     You mentioned the importance of thin films of transparent broadband oxides (TBO) in silicon solar cells as antireflection coatings (ARC). However, based on the results presented in Figure 7, the layers of Al2O3 – ZnO, Al2O3 – TiO2, and Al2O3 – ZrO2 appear to be less effective than the pure Al2O3 layer. Could you provide further insights into this observation?

In conclusion, I believe that addressing these comments will not only clarify certain aspects of your research but also enhance the overall quality of the manuscript.

Author Response

Dear Reviewer,

Thank you very much for your kind and interesting comments. We have expanded the manuscript with additional information according to your suggestions. All amendments were made and highlighted in the revised manuscript.

To answer point 2 specifically we believe that the ratios of Al/Zn, Al/Ti, and Al/Zr are consistent with the estimation as seen in Tables 1 and 2. The results confirming the consistency of the applied model with the percentages in the layers are presented in the tables. The calculated contents based on the EMA model differ from the values determined from the EDX measurement within the measurement error.

The discrepancy between the percentage of Al2O3 in Table 2 and the results from EDX that you mention are the result of the fact that the Table 2 shows the percentage of Al2O3 in the layer, while Table 1 shows the percentage of individual elements in the layer, hence the discrepancies.

Two-component layers can act as anti-reflective layers in multi-layer systems, where it is important to select a layer with a precisely defined refractive index. However, these are not the only applications of this type of oxide layers. Our goal was not to test this type of layers strictly for ARC applications, but to obtain information on how to obtain a specific refractive index of a layer made of two types of oxides. This knowledge will allow, for example, the use of this type of layers as components of Quarter Wavelength Optical Thickness (QWOT) systems.

Our task was to determine the dependence of transmission and refractive indices on the Al2O3 content in metallic oxide layers. We can conclude that the addition of Al2O3 increases the transmission and reduces the reflection in ZnO, TiO2 and ZrO2 oxides layers.

Reviewer 4 Report

Comments and Suggestions for Authors

This research investigates the optical properties of thin film alloys synthesized through atomic layer deposition (ALD), comprising broadband oxides such as ZnO, TiO2, and ZrO2, both in their pure form and doped with Al2O3. The ALD methodology involves precise deposition cycles, and various characterization techniques including scanning electron microscopy (SEM), energy-dispersive X-ray spectroscopy (EDX), and spectroscopic ellipsometry (SE) are utilized to analyze the films. Optical models such as the Sellmeier, extended Cauchy, and Bruggeman EMA models are employed to describe refractive index dispersion. The findings show that Al2O3 doping reduces the refractive index, allowing for the creation of films with tailored optical properties, and offering potential applications in anti-reflective coatings and filters.

Before recommending this work for publication, it is essential to highlight the following issues:

1. Insufficient Methodological Detail: The methodology section lacks specific details about the experimental setup, including information about the ALD process parameters, material sources, and measurement conditions. Providing a more comprehensive description of the experimental procedures would allow other researchers to replicate the work.

2. Data Interpretation: The paper presents extensive data, but there is a limited interpretation of the results. The significance of specific findings and their implications for the overall research objectives should be more thoroughly discussed to help readers understand the key takeaways.

3. Clarity and Structure: The paper's organization and clarity could be improved. The description of the research objectives, methods, results, and conclusions should be more logically structured and clearly articulated. Additionally, the paper should avoid excessive technical jargon and ensure that concepts are accessible to a broader scientific audience.

4. The approach of using the extended Cauchy model with additional parameters (D and R) in this work raises several concerns. Firstly, it adds complexity to the model, potentially requiring more data for accurate parameter fitting, which can be problematic if data is limited. Secondly, these added parameters lack a clear physical basis, making their interpretation challenging and potentially hindering insights into the underlying material behavior. Additionally, the risk of overfitting increases with the introduction of more parameters, which may lead to unreliable predictions when applied to new data or conditions. Furthermore, the model's generalizability to other materials or scenarios may be limited. Thus, careful consideration of the trade-offs between model complexity and data availability is necessary to justify its use in the study.

5. The described atomic layer deposition (ALD) methodology raises several issues: its complexity and the need for meticulous optimization of parameters can be time-consuming and challenging to reproduce consistently. The choice of precursors, particularly organometallic compounds, introduces safety and purity concerns. While interdiffusion is used for alloy composition homogenization, it may not always achieve the desired uniformity. The selection of substrates (Si and glass) limits generalizability, as different substrates can interact differently with deposited films. Variability in growth rates can lead to thickness deviations, and achieving reproducibility in ALD processes is often challenging. 

6. Related to Figure 1. A statistical outcome is needed. The main problem with this approach and subsequent modeling is that the SEM cross-section image, which is used to determine the thickness of the Al2O3-ZnO layer, may not provide an accurate representation of the layer's properties. SEM images can be affected by several factors, including the insulating properties of the layers and the difficulty in distinguishing the tested layer from any sputtered additional layers like gold or carbon. This can lead to inaccuracies in determining the layer thickness, which serves as a crucial input for modeling the results of ellipsometric measurements. Inaccurate layer thickness values can result in incorrect optical property predictions, potentially leading to unreliable conclusions about the alloy's optical behavior. To improve the accuracy of modeling, alternative methods for layer thickness measurement or characterization should be considered, especially when dealing with thin insulating layers that may pose challenges for SEM imaging.

Comments on the Quality of English Language

To enhance the English writing in this work, focus on clarity and conciseness, avoiding redundancy and ambiguous language. Maintain consistent terminology, correct grammar and punctuation errors, and prioritize the active voice. Thoroughly proofread the document for spelling and typographical mistakes, and ensure consistent formatting, including citations. Ensure logical flow between paragraphs and sections and strengthen the introduction and conclusion sections. Use figures and tables effectively, and consider the needs of both technical and non-specialist readers. Lastly, seek peer review for feedback on clarity, coherence, and technical accuracy to improve the overall quality of the paper.

Author Response

Dear Reviewer,

Thank you very much for your kind and interesting comments. We have expanded the manuscript with additional information according to your suggestions. All amendments were made and highlighted in the revised manuscript.

Referring specifically to point 4 of your review, the extended Cauchy model takes into account partial absorption in the layers, which is not present in the standard Cauchy model. Cauchy is an empirical model in which it is not assumed that the coefficients A, B, C and D have a physical meaning. Their selection is dictated by the best possible fit of the model to the theoretical results.

Round 2

Reviewer 2 Report

Comments and Suggestions for Authors

After the authors have revised and supplemented the text of the article, I believe that this work can be published in the Coatings MDPI journal.  In the current version, the novelty of the study is much better justified.

I have only one small comment regarding the main text of the article:

The formulas ZnMnO and ZnCoO (line 121) are not entirely correct. It is better to indicate them as Zn1-xMnxO and Zn1-xCoxO respectively

Author Response

Dear Reviewer,

Once again, thank you very much for all your comments, which allowed us to improve the content of our manuscript. The recommended change in the chemical formula (line 121) has been made and was highligted in the manuscript.

Reviewer 4 Report

Comments and Suggestions for Authors

Dear Authors,

Thank you for considering recommendations and suggestions. 

I recommend the publication of the revised version.

Author Response

Dear Reviewer,

Once again, thank you very much for all your comments, which allowed us to improve the content of our manuscript. Thank you for recommending our article for publication.